# Novel Epoxidized Brazil Nut Oil as a Promising Plasticizing Agent for PLA

**DOI:** 10.3390/polym15091997

**Published:** 2023-04-23

**Authors:** Aina Perez-Nakai, Alejandro Lerma-Canto, Ivan Dominguez-Candela, Jose Miguel Ferri, Vicent Fombuena

**Affiliations:** 1Technological Institute of Materials (ITM), Universitat Politècnica de València (UPV), Plaza Ferrándiz y Carbonell 1, 03801 Alcoy, Spain; aipena@epsa.upv.es (A.P.-N.); allercan@epsa.upv.es (A.L.-C.); vifombor@upv.es (V.F.); 2Instituto de Seguridad Industrial, Radiofísica y Medioambiental (ISIRYM), Universitat Politècnica de València (UPV), Plaza Ferrándiz y Carbonell s/n, 03801 Alcoy, Spain; ivdocan@doctor.upv.es

**Keywords:** epoxidized Brazil nut oil (EBNO), poly(lactic acid) (PLA), bio-plasticizers, mechanical properties, thermal properties, disintegration

## Abstract

This work evaluates for the first time the potential of an environmentally friendly plasticizer derived from epoxidized Brazil nut oil (EBNO) for biopolymers, such as poly(lactic acid) (PLA). EBNO was used due to its high epoxy content, reaching an oxirane oxygen content of 4.22% after 8 h of epoxidation for a peroxide/oil ratio of 2:1. Melt extrusion was used to plasticize PLA formulations with different EBNO contents in the range of 0–10 phr. The effects of different amounts of EBNO in the PLA matrix were studied by performing mechanical, thermal, thermomechanical, and morphological characterizations. The tensile test demonstrated the feasibility of EBNO as a plasticizer for PLA by increasing the elongation at break by 70.9% for the plasticized PLA with 7.5 phr of EBNO content in comparison to the unplasticized PLA. The field-emission scanning electron microscopy (FESEM) of the fractured surfaces from the impact tests showed an increase in porosity and roughness in the areas with EBNO addition, which was characteristic of ductile failure. In addition, a disintegration test was performed, and no influence on the PLA biodegradation process was observed. The overall results demonstrate the ability of EBNO to compete with other commercial plasticizers in improving the ductile properties of PLA.

## 1. Introduction

The use of plastics from fossil fuels has increased over the last century because of their wide range of applications, spanning from food packaging and bottles to furniture and household appliances, medical equipment, and construction materials. The increase is also due to their low production cost and characteristics, such as their balance between mechanical strength and lightness, their high resistance to degradation, and their electrical and thermal isolation capacity [1]. However, these materials come from non-renewable resources, and most of them cannot be easily degraded by solar radiation and microbial decomposers. They must be sorted in recycling processes, transferred to landfills, or put through incinerations for disposal. Nevertheless, it has been reported that only 10% of the 29.5 million tons of post-consumer plastic waste collected in Europe in 2020 was recycled [2]. The remaining 80% ended up in landfills, incinerations, and oceans, where they can last for decades or even centuries in these environments [3,4], accumulating and causing severe impacts on ecosystems. Plastic debris in oceans can cause serious injury, deformity, or intoxication to marine animals when consumed due to the addition of endocrine-disrupting compounds to the polymer matrix to improve flexibility or color properties [5].

To alleviate the environmental problems caused by petroleum-based plastics, several efforts have been performed to develop bioplastics, whether bio-based and/or biodegradable plastics. Those made from renewable resources, fully or partially, not only contribute to curbing environmental damage but also help to reduce dependence on fossil resources and, thus, the greenhouse effect. Some of the most promising bioplastics are (i) those obtained from biomass (cellulose, starch, protein, chitin, etc.) with partial modifications, such as thermoplastic starch (TPS); (ii) those obtained from the production of monomers by fermentation and polymerization, such as poly(lactic acid) (PLA), polybutylene succinate (PBS), and bio-polyethylene (bio-PE); and (iii) those obtained from microbial production, such as polyhydroxyalkanoates (PHA) and polyhydroxybutyrate (PHB) [6,7,8]. Currently, from the 367 million tons of plastic produced annually, only around 1% is bioplastics [1,5]. However, this market is expected to increase [4,9]. The packaging industry, which is the largest consumer of plastics and accounts for about 40% of plastic consumption, is also the largest consumer of bioplastics [2,4,9,10]. Non-biodegradable bio-based plastics, such as bio-polyethylene terephthalate (bio-PET) and bio-PE, are mainly used for rigid packaging. In contrast, biodegradable plastics, such as TPS and PLA, are used for flexible packaging [4].

One of the most commercially used biodegradable and bio-based polymers as an alternative to conventional petroleum-based polymers is PLA, which accounts for almost 19% the global bioplastic production [9]. This material has a competitive price, high transparency, low toxicity, and balanced mechanical properties comparable to other conventional commercial “commodity” polymers, such as polystyrene (PS), polypropylene (PP), polyethylene (PE), or poly(ethylene terephthalate) (PET) [11,12]. In addition, PLA offers relatively easy processing conditions, similar to those already mentioned, which make PLA a good option for a wide variety of applications [13]. Together, these features make this biopolymer a good alternative for applications in the packaging industry, automotives, textiles, and medical devices, among others [14]. Nevertheless, PLA presents a major drawback, namely its high intrinsic brittleness, which can be a disadvantage for some applications where flexibility is required [15]. Therefore, increasing its ductile properties (elongation at break, impact resistance, etc.), together with maintaining its property of being environmentally friendly, is one of the challenges for researchers in the field of biopolymers.

To improve PLA flexibility, there are two different typical ways. One possibility is to blend it with other polymers with high ductile properties, such as polyethylene glycol (PEG) [16], thermoplastic starch (TPS) [17,18], or polybutylene succinate-co-adipate (PBSA) [19,20], among others. However, although some improvement in toughness is achieved, most of these PLA-based blends show a certain degree of immiscibility, which is a major drawback to achieving synergistic effects [20]. For this reason, some authors have proposed modified vegetable oils (MVOs) as compatibilizing agents or as plasticizing additives themselves [1,13,21,22]. MVOs are a sustainable substitute for synthetic modifiers [23,24], while also being unharmful to human health due to their non-toxicity. In contrast, conventional epoxy resin plasticizers used in consumer products can generate the migration of toxic substances, such as bisphenol A (BPA) or phthalates [25,26], into the human body. In addition, most vegetable oils are derived from highly available renewable resources, which contribute positively to sustainable development, along with a cost-effective price.

There are two reactive sites in the fatty acids of VOs to increase their reactivity and compatibility for incorporation into a polymer matrix: double bonds and ester groups [27]. For this purpose, different techniques are available. Some of the most relevant are epoxidation [28,29], maleinization [30], acrylation, and hydroxylation. In particular, the epoxidation process has received attention due to its potential. This process consists of introducing epoxy groups (oxirane rings) into double bonds (Figure 1). Some commercially available epoxidized oils, such as epoxidized soybean oil (ESBO) and epoxidized linseed oil (ELO), have been shown to provide enhanced ductile properties to PLA [31,32]. Yu-Qiong Xu et al. reported an improvement of 63.32% in the elongation at break with a concentration of 9 wt% of ESBO [33]. Other authors have investigated other MVOs, such as epoxidized cottonseed oil (ECSO), epoxidized chia seed oil (ECO), and epoxidized palm oil (EPO), and obtained good results. For example, Carbonell-Verdu et al. developed PLA films with 10 wt% ECSO, showing an elongation at break result of 1100% and a remarkable increase in the impact energy absorption. In another study, Sempere-Torregrosa et al. achieved 107% elongation in modified PLA formulations with 7.5 wt% ECO. In addition, Awale et al. used EPO to improve the miscibility between PLA and starch and achieved very positive results, including a 600% elongation at break for the PLA/starch/10EPO formulation. Although these works cannot be compared with each other, improvements in ductility are observed in all of them.

In this work, the epoxidation of Brazil nut (*Bertholletia excelsa*) oil is carried out for first time. Traditionally, the use of this nut is for food. Nevertheless, in recent years, there has been a growing interest in this product since it contains a large amount of active molecules, with potential applications in multiple sectors, such as cosmetics, pharmaceuticals, and polymerics [34,35]. Brazil nut has a vegetable oil content between 60.8% and 72.5% [36,37]. It is an oil composed mostly of oleic and linoleic fatty acids, which gives it a high proportion of unsaturated fatty acids. However, despite the great potential for the use of Brazil nut oil in the field of biopolymers, the literature lacks such studies.

Brazil nut is produced along the Amazon and rarely comes from elsewhere. This Brazil nut monopoly is, therefore, a very important financial resource for people who produce and collect it. In fact, this industry currently supports thousands of rural and indigenous families, in addition to generating millions of dollars in exports [38]. Bolivia is the world’s leading producer and, together with Brazil and Peru, exports USD 123 million worth of Brazil nuts [39]. Furthermore, from an environmental point of view, Brazil nut cultivation brings great benefits to tropical forests since it is a large, ecologically valuable wild tree, and its cultivation ensures both its protection and that of the surrounding forest [40].

The aim of this study is, therefore, to test the potential of EBNO as a new bio-based plasticizer for PLA. EBNO is incorporated into the PLA matrix in different percentages to evaluate the mechanical and thermal properties. Lastly, the effect of EBNO on PLA degradation is analyzed using a disintegration test.

## 2. Materials and Methods

### 2.1. Materials

The PLA used in this study to improve its properties by plasticizing with epoxidized Brazil nut oil was commercial-grade Ingeo™ Biopolymer 3251D supplied by NatureWorks LLC (Minnetonka, MN, USA) in pellet form, with a density of 1240 kg·m^−3^ (ASTM D792) and a melt flow index of 80 g·10 min^−1^ when measured at 210 °C according to the ASTM D1238. This semi-crystalline PLA grade contains 1.4 wt% D-lactide. Its thermal transition temperature (T_g_) is between 55 and 60 °C, and its melting temperature (T_m_) is about 155–170 °C according to the ASTM D3418.

The Brazil nuts used in this work were supplied by FrutoSeco (Alicante, Spain). Brazil nut oil (BNO) was obtained by the cold mechanical extraction method using a pressing machine, model DL-ZYJ05, purchased from Nanchang Dulong Industrial Company (Weifang, China). The BNO obtained has a density of 916 kg·m^−3^ and an iodine value (IV) of 163.8 g I_2_·(100 g)^−1^, according to the ISO 1675 and ISO 3962, respectively. The epoxidation process of the BNO was carried out in situ with hydrogen peroxide (30% *v*/*v*), acetic acid (99.7), and sulfuric acid (97%) provided by Sigma-Aldrich (Madrid, Spain).

### 2.2. Epoxidation of Brazil Nut (Bertholletia excelsa) Oil

The epoxidation of the BNO was carried out in a 1000 mL three-neck round flask in a water bath with an immersion thermostat (Digiterm S-150 model from Selecta) with an accuracy of ±0.1 °C and a thermometer connected to a side neck to ensure the required temperature during the reaction (Figure 2). The central neck was used to place a stirrer, and the third neck was used to drop reagents and ensure an inert atmosphere.

The procedure was as described here: 232 g of BNO was incorporated into the flask and stirred constantly at a speed of 220 rpm until a temperature of 60 °C was reached. Afterward, 26.74 mL of acetic acid was added. Then, by dripping for 30 min to trigger the initiation of the reaction, 192.18 mL of hydrogen peroxide was added in a 2:1 ratio, together with 1.04 mL of sulfuric acid. Finally, the mixture was cooled at room temperature and cleaned by decantation and centrifugation.

The degree of epoxidation of the oil was determined according to the ISO 3961 using Equation (1):(1)$$IV=\frac{12.69\times c\left(V_{1}+V_{2}\right)}{m}$$
where *IV* is the iodine value in g I_2_·(100 g)^−1^ oil; c is the concentration in moles per liter of the sodium thiosulfate solution; *V*_1_ is the volume in mL of the thiosulphate sodium standard sample (mL); *V*_2_ is the volume in mL of the sodium thiosulphate used in the determination; and m is the mass of the weighted oil sample in grams.

On the other hand, the percentage of oxirane oxygen that has been introduced into the fatty acid chains is determined according to the ASTM D1652-97.

### 2.3. Manufacturing of PLA formulations with Epoxidized Brazil Nut Oil (EBNO)

Several PLA compositions were manufactured with different contents of EBNO (see Table 1). First, the PLA pellets were dried for 24 h at 40 °C in a drying oven (model Conterm from Selecta). The corresponding amount of the PLA pellets and EBNO were mixed manually until homogenized. The mixtures were then extruded using a co-rotating twin-screw extruder (D = 30 mm; L/D = 24:1) from DUPRA S. L. (Alicante, Spain), at a constant speed of 40 rpm. The temperature profile was set from 165 °C (feed zone) to 180 °C (die). The extruded materials obtained were cooled at room temperature and pelletized. Finally, the pellets were dried for 24 h at 40 °C to shape them into standard samples by an injection molding machine (model Meteor 270/75) from Mateu & Solé (Barcelona, Spain), at a temperature profile of 170 °C, 180 °C, 190 °C, and 200 °C from the feed section to the injection nozzle. The injection pressure was set at 60 bar, and the compaction pressure was set at 55 bar for 20 s (compaction time). The injection speed was 20 cm^3^/s, and the mold temperature was set at 25 °C.

### 2.4. Mechanical Characterization of PLA Formulations Plasticized with EBNO

Standard tensile, impact, and hardness tests were conducted to determine the mechanical properties of the different PLA compositions with different EBNO contents. The tensile test was carried out according to the ISO 527 with an Ibertest ELIB 30 universal testing machine from SAE Ibertest (Madrid, Spain). The specimen geometry used for the tensile test was a 1A type. The test was performed at a crosshead speed of 10 mm·min^−1^ with a load cell of 5 kN. A minimum of 5 different samples of each material were tested to obtain the average values of the elongation at break and the tensile strength. An axial extensometer from Ibertest was used to obtain the values of Young’s modulus.

The impact resistance by energy absorption was measured on a 6 J Charpy’s pendulum from Metrotec S.A. (San Sebastian, Spain). The samples used were rectangular, with a dimension of 80 mm × 10 mm × 4 mm and without notches, following the guidelines of the ISO 179. Finally, hardness was tested with a Shore D hardness tester, model 673-D from J. Bot S.A. (Barcelona, Spain), according to the ISO 868. The results obtained were an average of a minimum of 5 different samples of each material.

### 2.5. Thermal Analysis of PLA Formulations Plasticized with EBNO

The thermal properties of the PLA formulations with EBNO were evaluated by using differential scanning calorimetry (DSC) and thermogravimetric analysis (TGA).

The DSC was carried out on a Mettler Toledo DSC 821 (Schwerzenbach, Switzerland). The samples, with an average weight of 6–8 mg, were tested under nitrogen atmosphere (flow rate of 66 mL·min^−1^) using a thermal program consisting of a 1st heating to move the thermal history from 30 °C to 300 °C at 10 °C·min^−1^, followed by cooling to 30 °C at 10 °C·min^−1^, and a final 2nd heating from 30 °C to 300 °C at 10 °C·min^−1^. From this second heating, thermal transitions, such as the glass transition temperature (T_g_), cold crystallization (T_cc_), and melt peak temperature (T_m_), were determined. The crystallinity percentage (*X_c_*) was determined by using Equation (2):(2)$$X_{c\;}\left(\%\right)=\frac{\Delta H_{m}-\Delta H_{c}}{w\Delta H_{m}^{o}}\times 100$$
where *∆H_m_* and *∆H_c_* are the melting enthalpy and the cold crystallization enthalpy (J·g^−1^), respectively; *w* is the mass fraction of the material; and ∆$H_{m}^{o}$ is the theoretical melt enthalpy for a 100% crystalline PLA structure, which is assumed to be 93 J·g^−1^ [41]. Three analyses were performed for each formulation, and the curve of the intermediate values was plotted. The samples were cut from the cross section of the impact specimens.

The TGA tests were carried out on a TGA/SDTA 851 from Mettler Toledo (Schwerzenbach, Switzerland). The samples, with an average weight between 7 and 10 mg, were subjected to a heating program from 30 °C to 700 °C at a constant heating rate of 20 °C·min^−1^ under nitrogen atmosphere (flow rate of 66 mL·min^−1^). The temperature (T_5%_) at which the weight loss was 5% and the temperature of maximum degradation (T_max_) were calculated to evaluate the thermal stability of the formulations of PLA with EBNO.

### 2.6. Thermomechanical Characterization of PLA Formulations Plasticized with EBNO

The thermomechanical properties of the PLA formulations with EBNO were evaluated by using dynamic thermomechanical analysis (DMTA) and heat deflection temperature (HDT).

The DMTA was carried out to analyze changes in the storage modulus (G’) and damping factor. The test was performed with an AR G2 oscillatory rheometer, from TA Instruments (New Castle, EE.UU), equipped with a clamp attachment for solid samples. The samples were tested in the torsion mode and subjected to a thermal program from 30 °C to 130 °C at a heating rate of 2 °C·min^−1^. The frequency was set at 1 Hz, and the maximum deformation (γ) was 0.1%. The T_g_ was determined from the tan δ peak results.

For the HDT test, a DEFLEX 678-A2 station (Metrotec S.A., San Sebastián, Spain) was used. The values were obtained following the guidelines of the ISO 75 at a constant heating rate of 120 °C·h^−1^ and a constant load of 296 g.

### 2.7. Morphology of PLA Plasticized with EBNO

The cross-sectional areas of the fractured samples in the impact test were observed by using field-emission scanning electron microscopy (FESEM), model ZEISS ULTRA 55 from Oxford Instruments (Tubney Woods, Abingdon, Oxfordshire, UK). The samples were first coated with a gold layer under vacuum conditions to increase their electrical conductivity by employing an EM MED020 sputter coater from Leica Microsystems (Wetzlar, Germany). All fractured surfaces of the samples were observed using an accelerating voltage of 2 kV.

### 2.8. Degree of Disintegration under Composting Conditions of PLA Formulations Plasticized with EBNO

The disintegration test was conducted in aerobic conditions according to the ISO 20200 at a temperature of 58 °C and a relative humidity of 55%, using a 300 mm × 200 mm × 100 mm synthetic compost reactor. The disintegration samples were manufactured from the pellets obtained from the extrusion process using the hot plate technology. For this purpose, the necessary amount of material to fill the mold (100 mm × 100 mm × 1 mm steel frame) was placed, and the plates were heated to 190 °C. Once the temperature was reached, the frame with the material was placed inside the plates and gradually pressed until a pressure of 5 bar was reached. When the pressure stabilized, it means that all of the material had melted and a sheet had been generated, which would later be cut into squares of 25 mm × 25 mm × 1 mm. Five sheet samples for each compound were dried at 40 °C for 24 h and then placed in a carrier bag and buried in controlled soil. To evaluate the degree of disintegration and the disintegration process under the compost conditions, control days (4, 9, 14, 21, and 28 days) were established to unearth a sample of each formulation. These removed samples were washed with distilled water and dried at 45 °C for 24 h. Finally, the extracted samples were weighed, and their disintegration percentage was calculated using Equation (3) from the ISO 20200:(3)$$D=\frac{m_{i}-m_{r}}{m_{i}}\times 100$$
where *m_i_* refers to the initial mass of the sample prior to the test, and *m_r_* is the weight of the sample extracted from the compost soil on different control days. Furthermore, optical images were taken of each sample to assess the disintegration process qualitatively.

## 3. Results

### 3.1. Syntesis of Epoxidized Brazil Nut Oil

The evolution of the IV along the time during the epoxidation process of the BNO is presented in Figure 3. Initially, the IV of the oil is 102.4 g I_2_·(100 g)^−1^, similar to the value reported by Pena Muniz et al., which is 95 g I_2_·(100 g)^−1^ [42]. It can be observed how the value decreases significantly after the first two hours to 52.2 g I_2_·(100 g)^−1^ due to the high availability of unsaturated fatty acids, indicating a conversion of 49.02% of double bonds, meaning that epoxidation is occurring. This value keeps decreasing and, when the reaction reaches 8 h, the IV is 11.6 g I_2_·(100 g)^−1^, representing a conversion of 88.67% of double bonds. Carbonell-Verdu et al. showed a similar result with cottonseed oil epoxidation, obtaining a double bond conversion of 98.3% [43]. Regarding the oxirane oxygen index in the Brazil nut oil, it is theoretically 6.11% by calculating the average values of its fatty acids. As it can be seen, the oxirane oxygen index reaches 4.22% after 8 h, which represents a performance of 69% (for a peroxide/oil ratio of 2:1). Nevertheless, it is important to take into account some other parallel reactions that may occur, such as epoxy homopolymerization, which do not contribute to the increase in oxirane oxygen but can lead to discrepancies in the conversion of the IV [43].

### 3.2. Effects of EBNO on Mechanical Properties of Plasticized PLA Formulations

Since PLA is a brittle material, different percentages of EBNO were used as a plasticizer to improve its mechanical properties. The tensile strength, elongation at break, and tensile modulus values of the studied materials are presented in Figure 4. It is observed that, by incorporating EBNO into the PLA, the tensile strength decreases strongly concerning the neat PLA. By incorporating only 2.5 phr of EBNO, the tensile strength decreases from 55.9 ± 1.4 MPa to 33.0 ± 2 MPa for that formulation. It is a strong reduction in the strength representative of a plasticized polymeric material. Therefore, the ductile properties increase as the amount of EBNO incorporated increases, achieving an elongation at break of 22.3% for the formulation with 7.5 phr of EBNO, which is 70.9% more compared to the unplasticized PLA. Sempere-Torregrosa et al. showed the same trend when mixing PLA/25PHB with 7.5 phr of epoxidized corn oil (ECO). Specifically, in their study, the elongation at break was increased by 107% [44]. The tensile strength decreased from 61.6 ± 2.7 MPa for the unplasticized PLA to 27.8 ± 1.2 MPa [44]. Therefore, the trend is the same in both studies, although in the present study, the increase in ductility is less. However, in both studies, the elongation at break shows a maximum at different epoxidized oil contents. This is due to oil saturations, as shown in the FESEM images. Although the materials with a higher epoxidized oil content have greater ductility, saturations do not allow the elongation at break to increase, causing premature breakage. Garcia-Garcia et al. showed similar trends using 5 wt% of epoxidized karanja oil (EKO) in PLA [45].

In addition, the Young’s modulus usually decreases to a greater or lesser degree in the presence of a plasticizing agent. In the current case, we passed from a Young’s modulus of 2556.4 ± 167.6 MPa to 2274.6 ± 192.1 Mpa, which represents a reduction of 11.0% concerning the Young’s modulus of the PLA. The trends of the tensile strength, elongation at break, and tensile modulus show that EBNO acts as a plasticizing agent, as demonstrated in other works [45,46].

On the other hand, the impact absorption energy increases (Figure 5) but only up to the formulation with 2.5 phr of EBNO. The saturation of EBNO in the PLA matrix from this concentration is responsible for this decrease, as shown inthe FESEM section. When the strain rate is controlled and is low (tensile tests), the saturation does not affect the result too much since the polymer chains are given time to slide over each other, allowing a higher elongation at break. However, when it comes to deformation caused by impact (high speed), the material does not have time to deform and break, thereby absorbing certain energy. In addition, oil saturations act as stress concentrators, resulting in earlier breakage and less energy absorption.

In terms of hardness, it is observed that all PLA formulations plasticized with EBNO have a lower hardness. Specifically, it decreases from 82.2 for the unplasticized PLA to almost 80 Shore D for the PLA formulations with a higher EBNO content. This decrease is indicative of the plasticizing effect of EBNO. The same trend can be observed in other works [13,22].

### 3.3. Effects of EBNO on Thermal Properties of Plasticized PLA Formulations

The thermal stability of the PLA and the plasticized formulations with different contents of EBNO was assessed by thermogravimetry analysis (TGA). Figure 6A shows the weight loss versus temperature curves for each sample (TG), and Figure 6B shows their corresponding first-derivative curves (DTG). Table 2 shows the main thermal parameters, such as the degradation onset temperature (T_5%_), which indicates the temperature at which a 5% weight loss occurs, and the maximum degradation rate (T_max_), which corresponds to the peak of the first-derivative curve. The unplasticized PLA possesses a thermal stability value with a T_5%_ of 329.9 °C and a T_max_ of 375.3 °C. However, the addition of certain amounts of EBNO to the PLA causes a slight decrease in the T_5%_ and T_max_ values. These values are lower when the quantity is larger. Specifically, the addition of 10 phr of EBNO results in a T_5%_ reduction of 14 °C and a reduction of 7 °C for T_max_. This might be due to the fact that the oil degrades at lower temperatures than the PLA itself. This leads to a greater mass loss when there is a greater amount of EBNO contained in the formulation. It should be noted that part of the oil does not interact with the PLA and remains isolated (saturated). Therefore, the higher the oil content, the higher the saturated oil content and the higher the tendency for T_5%_ and T_max_ to decrease more.

A similar tendency was observed by Jama Awale et al. [47] for PLA/starch blends compatibilized with epoxidized palm oil (EPO). The addition of EPO into the blends resulted in a reduction in T_5%_; however, the T_max_ increased slightly.

Regarding the DSC results, Table 2 also shows the main thermal properties, and Figure 7 shows the calorimetric graph of the unplasticized PLA and the plasticized formula-tions. The unplasticized PLA has a glass transition temperature (T_g_) of 61.6 °C, a cold crystallization temperature (T_cc_) of 124.8 °C, and a melting temperature (T_m_) of 166.8 °C. EBNO reduces the T_g_ of the PLA, which indicates an increase in the mobility of the polymer chains at lower temperatures, evidencing the plasticizing effect of EBNO. For example, the specific case of the PLA plasticized with 7.5 phr of EBNO shows a decrease of 3.7 °C. The same plasticizing effect was observed by Dominguez-Candela et al. [27], who employed epoxidized chia seed oil (ECSO) in PLA and observed a slight decrease in T_g_ and a little variation in T_m_. In the present work, no significant variations were observed in T_m_ for the formulations with EBNO. This could be due to the fact that the decrease that should be registered might be compensated by an increase that could have resulted by achieving a greater degree of crystallinity.

With respect to crystallinity, it is observed that the unplasticized PLA has a crystallinity of 5.3%. The incorporation of EBNO leads to an important increase with respect to PLA. In fact, for the formulation containing 7.5 or 10 phr of EBNO, a significant increase in crystallinity of more than 400% is achieved. The T_cc_ peak is observed at lower temperatures with the incorporation of EBNO, specifically decreasing from 124.8 °C (unplasticized PLA) to 98.9 °C for the PLA with 7.5 phr of EBNO. This is due to the increase in the chain mobility promoted by the EBNO. The same trend is shown in the work of Dominguez-Candela, where it starts from a crystallinity degree of 7.5% for the neat PLA to 11.5% for the PLA with 10 phr of ECSO. Although they used a PLA grade called amorphous (2003D) because of its low crystallinity, what is clear is that they also experienced an increase of 53.3% in crystallinity, and this may be the reason why the T_m_ does not decrease with the use of a plasticizer.

### 3.4. Effects of EBNO on Thermomechanical Properties of Plasticized PLA Formulations

Following the thermomechanical characterization of the samples, the first test carried out was the heat deflection temperature (HDT) test. Table 3 shows the average results obtained for the different PLA samples, both unplasticized and plasticized with different percentages of EBNO. The trend obtained is the same as previously reported for the evolution of T_g_ measured by DSC. The highest value was obtained with the unplasticized PLA sample, with an HDT of 59.2 °C. After the addition of EBNO, the plasticizing effect provides to the PLA matrix results in a decrease in the HDT values. The sample with the lowest value obtained is the PLA with 7.5 phr of EBNO, in the same way that occurs with the T_g_ value or even, referring to the mechanical tests, the sample with the highest elongation at break. The decrease in HDT is 5.4% compared to the unplasticized PLA sample. A higher addition of EBNO (10 phr) does not seem to increase the plasticity of the samples, as detected in the mechanical and thermal tests, and in this HDT test. The same plasticizing effect was observed by García-Campo et al. [48] using ESBO, where a decrease of 15% in the HDT was achieved. It is evident that no significant drop in HDT is observed with the addition of EBNO, as expected with the addition of a plasticizer. The effect of the increased crystallinity may lead to a stiffening effect and increased resistance to its deformation or softening at higher temperatures than it should show if there were no increase in crystallinity.

The dynamic mechanical response of the processed materials was evaluated using the storage modulus (G’) and the damping factor (tan δ). Figure 8 shows the graphical evolution of both. First, regarding the storage modulus (G’) in Figure 8A, it is worth noting a first temperature range between 30 °C and 55 °C where the samples remain stable, without significant decreases. Therefore, this is the temperature range where the PLA and the EBNO-plasticized PLA materials remain thermally and mechanically stable. The first sharp change in the G’, between 55 °C and 70 °C, is attributable to the T_g_ of the PLA, as demonstrated by the previous DSC test or as some authors have shown [49]. The second detectable transition occurs between 70 °C and 90 °C and is attributable to the cold crystallization process. In this stage, there is a rearrangement of molecules, causing a greater degree of packing, and consequently, a slight increase in thermomechanical properties. Regarding the addition of EBNO, it does not seem to make a big difference in the evolution of the G’ at low temperatures, although after the T_g_, its addition causes a greater loss of thermomechanical properties. EBNO can act by increasing the free volume between PLA molecules, reducing interaction and, as a result, causing a plasticizing effect prior to the saturation of the matrix.

The plasticizing effect promoted by EBNO in the PLA matrix is more easily quantifiable in the evolution graph of tan δ versus temperature, as shown in Figure 8B. The unplasticized PLA samples have a T_g_ around 63.8 °C, while the magnitude of their peak is reduced. This is due to the reduced molecular mobility of the unplasticized PLA. The samples with 7.5 phr and 10 phr of EBNO result in lower T_g_ values, specifically 62.1 °C and 60.0 °C, respectively. The same trend had been recorded through the DSC test and similar studies. Dominguez-Candela et al. [27] demonstrated the same behavior using epoxidized chia seed oil as a plasticizer. On the other hand, other authors, such as Chieng et al. [50], demonstrated that an increase in plasticizer content leads to higher magnitude peaks, as a consequence of greater molecular mobility.

### 3.5. Morphology of PLA Formulations with EBNO

The plasticization effect caused by the addition of EBNO also has an influence on the morphology of the fracture surfaces of the samples. Figure 9 shows the images obtained by FESEM at 1000× magnification. The unplasticized PLA, as shown in Figure 9A, is characterized by a fairly smooth surface, without any features of plastic deformation. This is due to its brittleness, as demonstrated by the Charpy impact test (21.4 kJ·m^−2^). The addition of EBNO as a plasticizer, even in low proportions, as shown in Figure 9B, influences the toughness of the sample and its morphology. The roughness of the fracture surface is slightly higher, and small hollow microspheres caused by the addition of the oil can be seen (highlighted by red arrows). As a result of the addition of EBNO, the toughness of the sample increases by 60%. Figure 9C, obtained on a sample with 5 phr of EBNO, shows a higher density of hollow microspheres caused by the larger amount of EBNO. The high density and size of these microspheres may be one of the reasons why toughness is not increased when compared to the unplasticized PLA sample. Higher additions of EBNO, as shown in Figure 9D,E, clearly show fracture surfaces that are characteristic of ductile failure, with increased roughness and topographic changes on the surface. Microspheres caused by the addition of EBNO appear again (red arrows), but signs of saturation, such as the presence of filaments (yellow arrows), can also be observed. In the sample with 10 phr of EBNO, the density of filaments increases. These signs of saturation of the polymer matrix due to the high amount of plasticizer introduced can cause a slight phase separation and a lack of miscibility [45]. This leads to a lack of improvement in the ductile properties of PLA, as previously demonstrated in the mechanical tests. Similar results have been obtained in different studies using epoxidized cottonseed oil and maleinized linseed oil [13,51].

### 3.6. Disintegration under Composting Conditions of PLA Formulations

A key aspect in the development of new plasticizers for biopolymers, such as PLA, is their non-influence on the biodegradation process. For this purpose, the disintegration study of the different samples was carried out under composting conditions. Figure 10 and Figure 11 show the visual evolution of the samples after different burial times in the compost and the evolution of their mass loss. In this type of study, as different authors have shown, the crystallinity of the sample is a very important aspect in the disintegration rate [52]. Biodegradation processes that are carried out by different microorganisms present in the compost act more rapidly in the amorphous domains of biopolymers [53,54]. Therefore, a key aspect of this study is to demonstrate whether EBNO, which has been shown in the DSC study to increase the crystallinity of PLA, negatively influences the composting disintegration process.

As shown in Figure 10 and Figure 11, during the first four days of incubation under thermophilic composting conditions, no significant visual differences are observed. The only difference is an increase in opacity due to the higher crystallinity of the samples as a result of the proximity of the PLA’s T_g_ (61.6 °C) to the thermophilic temperature of the test (58 °C). During this period, no weight loss is observed in the samples. After nine days of incubation, the first weight losses are detected, being around 15–25 wt%. The samples dug up for the first time have disintegrated into relatively large pieces. The breakage of the samples is less evident in the samples with 7.5 and 10 phr of EBNO. These samples have a *X_c_* of 26–27%, which is much higher than the 5.3% of the neat PLA. This is the reason why the composting disintegration process is less evident. However, despite this increase in crystallinity provided by the EBNO, from day 15 of incubation, the visual appearance of all samples tends to homogenize and becomes very similar between samples, such as between the unplasticized PLA and the PLA with 10 phr EBNO. The weight loss ratio remains at a constant rate from day 9, and all samples reach a degradation ratio of 90% after 27 days of incubation. According to the ISO 20200 standard, a polymer can be considered biodegradable by composting if it degrades by 90% in less than six months. Therefore, based on the results, it can be stated that the addition of EBNO does not influence the composting biodegradation rate, as a 90% degradation is achieved in less than 30 days. Some authors had obtained a certain delay in the degradation rate as a consequence of the addition of chemically modified oils as plasticizers [55,56,57]. However, EBNO does not affect the degradation rate, being able to plasticize PLA without affecting its biodegradation under composting conditions.

## 4. Conclusions

For the first time, epoxidized Brazil nut oil (EBNO) has been developed. This research work evaluates EBNO as a new potential environmentally friendly plasticizer. It was introduced into the PLA matrix in different amounts to study its effect on the properties of this biopolymer, which is characterized by a low intrinsic ductility. By adding only 2.5 phr of EBNO, an improvement in the brittleness of the PLA is already observed, reaching a maximum increase in the elongation at break with 7.5 phr of EBNO (70.9% more in comparison to the unplasticized PLA). Then, in the PLA with 10 phr of EBNO, this value decreases due to oil saturation in the matrix. At the same time, the tensile strength and the Young’s modulus decrease, indicating a plasticizing effect of the polymeric material. As for the thermal parameters, when EBNO is added, there is no significant effect on the T_m_; however, the T_g_ decreases, with the lowest loss of 3.7 °C when 7.5 phr of EBNO is added, which is in full agreement with the performance obtained in the test on mechanical characterization. On the other hand, crystallinity increases significantly, with an increase of more than 400%, as the plasticizer increases the free volume, thereby improving the mobility of the PLA chains. On the other hand, the FESEM analysis revealed how the smooth morphology of the fractured surface of the unplasticized PLA changes to rough and porous with the addition of EBNO. As the amount of EBNO increases, signs of saturation appear, which is in full agreement with the mechanical and thermal results. Finally, the disintegration test showed that the addition of EBNO does not delay the biodegradability process of the PLA. Therefore, EBNO shows great potential as a bio-plasticizer to decrease the brittleness of PLA, obtaining results similar to those of other commercial plasticizers.

## Figures and Tables

**Figure 1 polymers-15-01997-f001:**
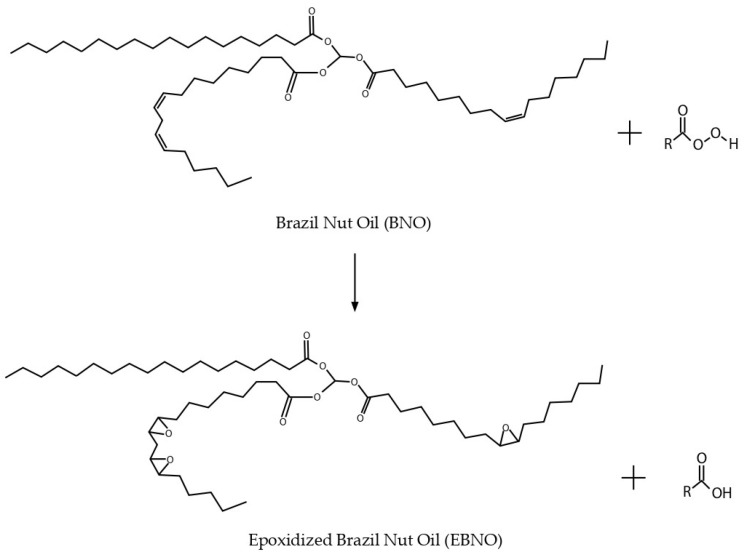
Schematic representation of epoxidation of Brazil nut oil.

**Figure 2 polymers-15-01997-f002:**
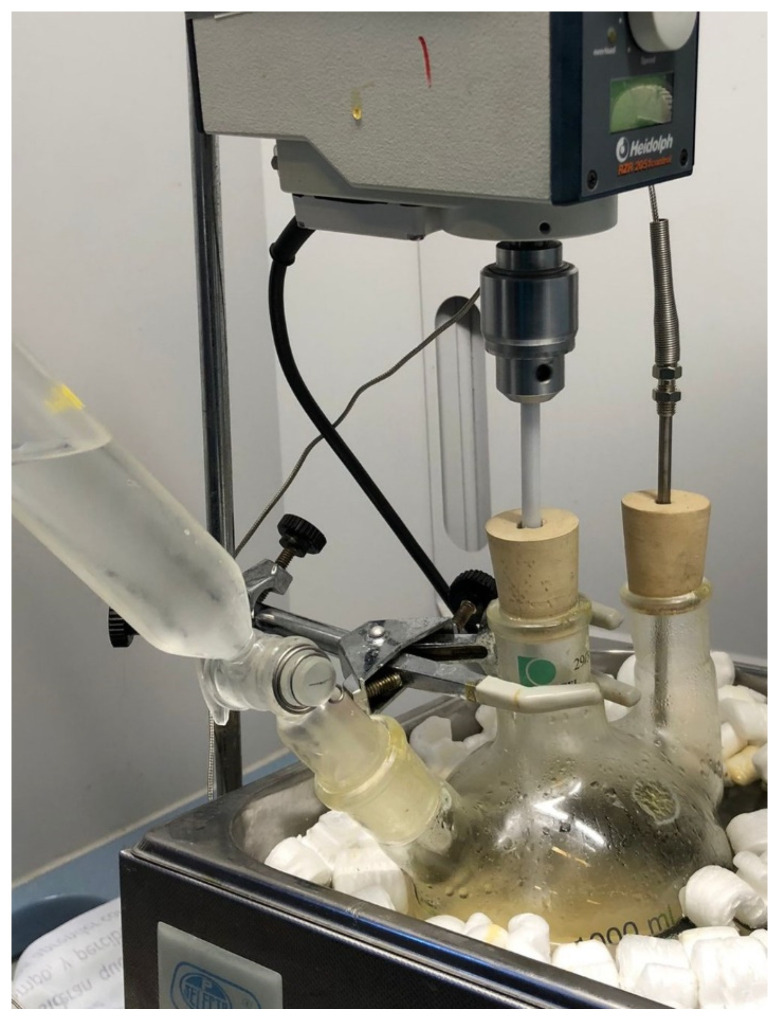
Schematic representation of the epoxidation process of Brazil nut oil.

**Figure 3 polymers-15-01997-f003:**
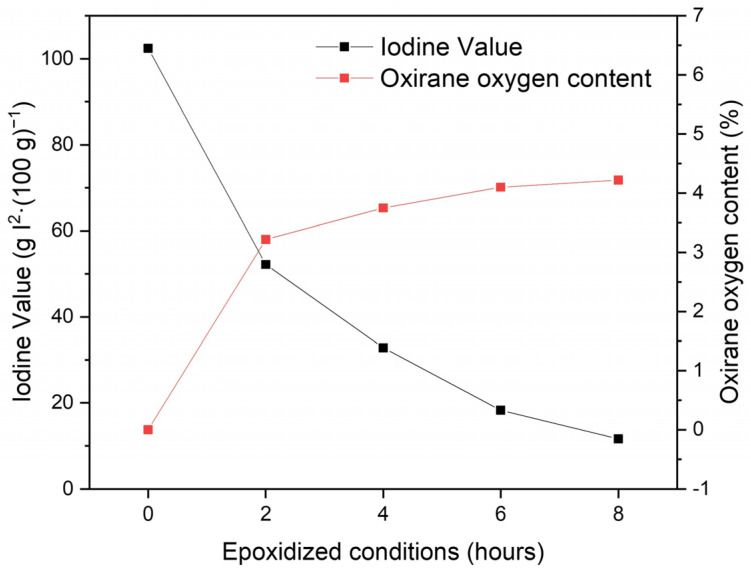
Effect of time on the efficiency of epoxidation of BNO.

**Figure 4 polymers-15-01997-f004:**
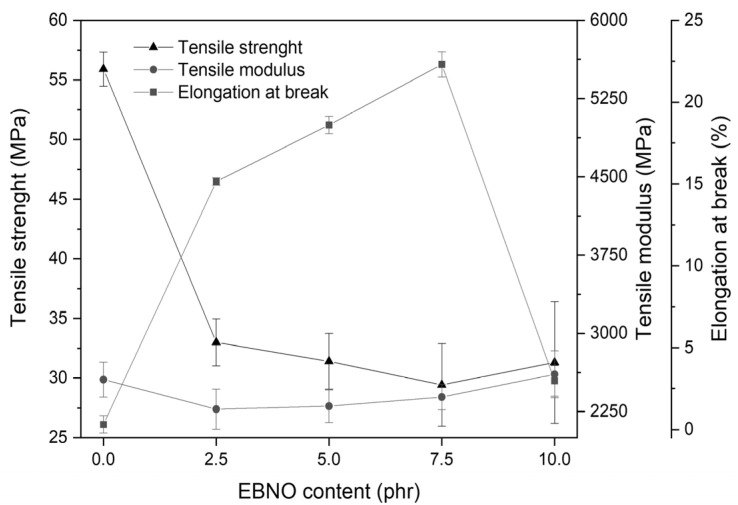
Effects of epoxidized Brazil nut oil content on the tensile strength, elongation at break, and tensile modulus of the plasticized PLA formulations.

**Figure 5 polymers-15-01997-f005:**
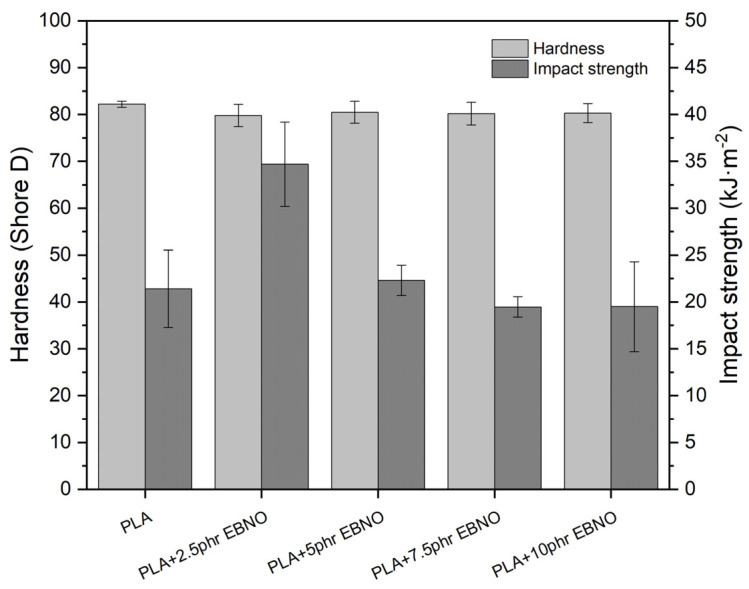
Effects of epoxidized Brazil nut oil content on the Shore D hardness and impact strength of the plasticized PLA formulations.

**Figure 6 polymers-15-01997-f006:**
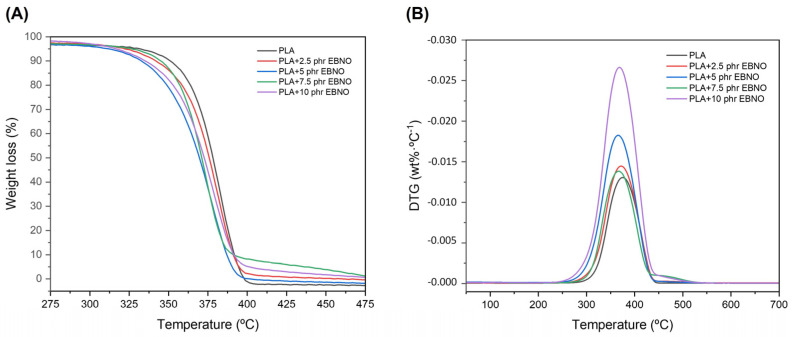
TGA (**A**) and DTG (**B**) of unplasticized PLA and PLA plasticized with different contents of EBNO.

**Figure 7 polymers-15-01997-f007:**
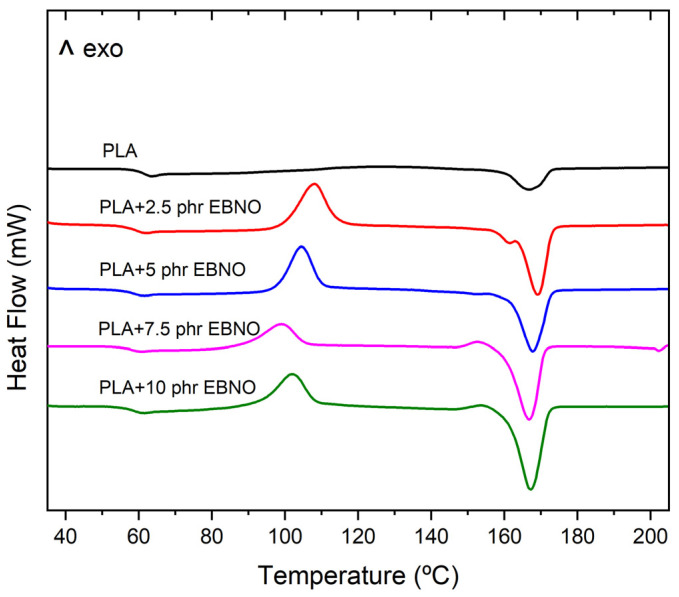
Dynamic DSC curve of the unplasticized PLA and plasticized PLA with different amounts of EBNO.

**Figure 8 polymers-15-01997-f008:**
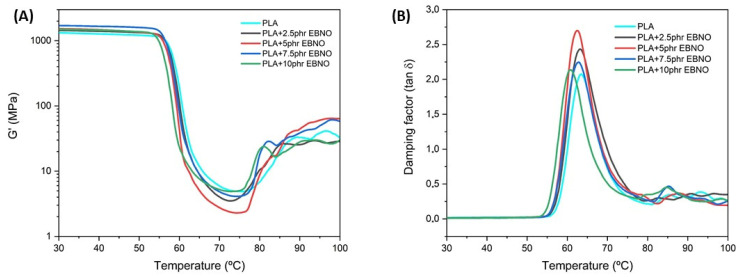
Plot evolution of dynamic mechanical thermal analysis (DMTA) of the neat PLA and the plasticized formulations with different EBNO contents: (**A**) storage modulus (G’) and (**B**) damping factor, tan (δ).

**Figure 9 polymers-15-01997-f009:**
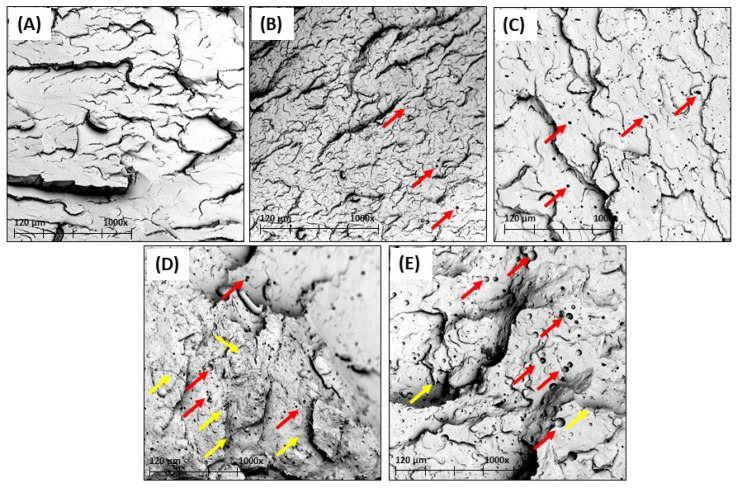
FESEM images (magnifications ×1000) of the fracture samples from the impact test on (**A**) PLA, (**B**) PLA + 2.5 phr EBNO, (**C**) PLA + 5 phr EBNO, (**D**) PLA + 7.5 phr EBNO, and (**E**) PLA + 10 phr EBNO.

**Figure 10 polymers-15-01997-f010:**
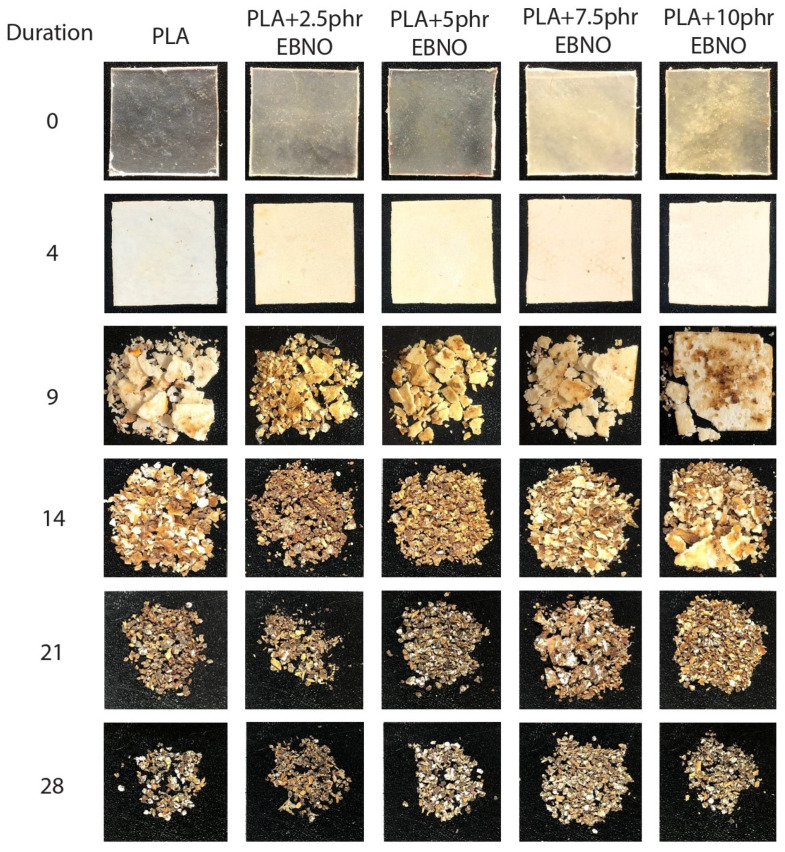
Visual appearance of disintegration of the unplasticized PLA and plasticized PLA samples under composting conditions.

**Figure 11 polymers-15-01997-f011:**
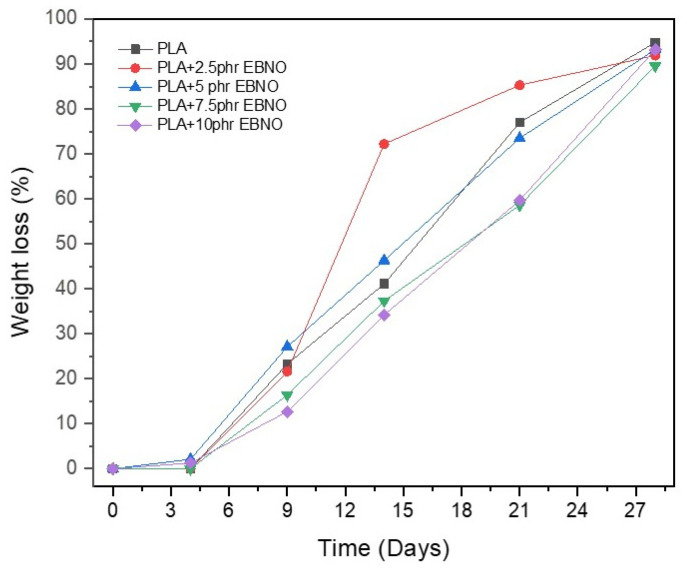
Plot evolution of the weight loss recorded during the composting process.

**Table 1 polymers-15-01997-t001:** Composition of plasticized PLA with different contents (phr) of EBNO and the labeling of the formulations.

Parts Per Hundred Resin (phr)		
Reference	PLA	Epoxidized Brazil Nut Oil (EBNO)
PLA	100	0
PLA + 2.5 phr EBNO	100	2.5
PLA + 5 phr EBNO	100	5
PLA + 7.5 phr EBNO	100	7.5
PLA + 10 phr EBNO	100	10

**Table 2 polymers-15-01997-t002:** Summary of the main thermal parameters of the unplasticized PLA and plasticized formulations with different EBNO contents.

Samples	TGA Parameters		DSC Parameters (2nd Heating)					
	T_5%_ (°C)	T_max_ (°C)	T_g_ (°C)	T_cc_ (°C)	*ΔH_c_* (J g^−1^)	T_m_ (°C)	*ΔH_m_* (J g^−1^)	X_PLA_ (%)
PLA	329.9	375.3	61.6	124.8	10.1	166.8	15.0	5.3
PLA + 2.5 phr EBNO	320.6	370.7	59.6	108.1	32.2	169.1	38.2	6.6
PLA + 5 phr EBNO	311.3	368.3	58.9	104.4	30.3	167.9	40.9	12.0
PLA + 7.5 phr EBNO	325.3	366.0	57.9	98.9	21.5	166.8	44.9	27.1
PLA + 10 phr EBNO	315.9	368.3	59.8	101.9	22.3	167.3	45.1	26.7

**Table 3 polymers-15-01997-t003:** Summary of the evolution of HDT of the unplasticized PLA and plasticized formulations with different EBNO contents.

Samples	HDT (°C)
PLA	59.2 ± 0.5
PLA + 2.5 phr EBNO	58.8 ± 0.4
PLA + 5 phr EBNO	58 ± 0.6
PLA + 7.5 phr EBNO	57.8 ± 0.5
PLA + 10 phr EBNO	57 ± 0.4

## Data Availability

Not applicable.

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
