# Peer review of "Novel Epoxidized Brazil Nut Oil as a Promising Plasticizing Agent for PLA"

_polymers, 2023, doi:10.3390/polym15091997_

Round 1

Reviewer 1 Report

The authors of the publication covered a very interesting topic in their paper. For the first time, epoxidized Brazil nut oil was developed and then introduced into the PLA matrix in various amounts. The present study was aimed at determining the effect of Brazil nut oil on the properties of the PLA biopolymer. From the descriptions in the work presented for review, it can be seen that such modification of the PLA structure has a beneficial effect on its properties. In summary, the work is very interesting and innovative. I have no major comments, while it would be advisable to include an illustrative drawing or photo of the apparatus for EBNO epoxidation. 

Reviewer 2 Report

The goal of the article is to evaluate the possibilities of an environmentally friendly epoxidized Brazil nut oil (EBNO) as a plasticizer for poly(lactic acid) (PLA) biopolymer. The importance and actuality of the topic are properly explained in the introduction. The article is well organised. However, there are a lot of formal misstatements and mistakes which should be corrected. These are missing information, clarification of some explanations and the level of English. Unfortunately, there are also more serious errors that have to be resolved. These are mainly insufficient or incorrect evaluation of results (meaning of standard deviation) or relevance of the decomposition experiment.  If these mistakes are corrected and the following questions are answered/ implemented in the manuscript, the paper can be recommended for publication.

Point 1: The quality of the English Language should be improved. For example, the explanation of the degradation of non-renewable resources in lines 34 - 36 is confusing. The phrase “PLA used for mixing” in line 125 does not describe the point of your research. The goal is to improve the properties of PLA. Therefore, use better explanations, such as plasticized/modified PLA. These are only some examples. There are many more of these misstatements in the manuscript. Therefore, do a detailed correction of the entire manuscript in terms of meaning and level of English.

Point 2: You are writing in line 97 that some authors have investigated other MVOs, such as epoxidized cottonseed oils (ECSO), etc. You should be more specific. Who investigated it? Which results were achieved? This part of the introduction is for your work crucial. Consequently, you should show more examples here.

Point 3: You should mention in line 126 the standards used to determine density and melt flow index (melt flow index 23 g/10 min, ISO 1133-A). Could you mention other important parameters of PLA such as melting point, glass transition temperature, stereochemical purity that are for processing and properties of PLA composites crucial?   

Point 4: Could you specify the temperature regulator thermometer (type, company, accuracy, etc.)? (line 137)

Point 5: There was an applied inert atmosphere within the epoxidation of Brazil Nut? Which one and how was it achieved? (line 139)

Point 6: Specify equipment used for drying (name, company, etc.) (line 158)

Point 7: Add the name of a used extruder and injection machine. (lines 160 - 164)

Point 8: Which samples were produced for evaluation of the mechanical and thermal properties of modified PLA? Was it 1A type standard tensile test specimens or a different one? Add this information to the manuscript (line 163). Add also process parameters used in the injection process such as injection and holding pressure, the temperature of mold, etc. Especially mold temperature could significantly influence the structure of PLA and its mechanical properties.

Point 9: Why do you not count with primary crystallization enthalpy (line 195) in equation (2) when from DSC curves is evident in some material combinations? Do you count with a level of primary crystallization in the results of crystallinity degree? This could change the results of crystallinity degree.

Point 10: You described in line 228 samples used to determine the degree of disintegration under the composting condition as 25 x 25 x 1 mm3. Unit mm3 for size? The same is mistake is in line 227. It is better to say samples with length, thickness, etc.  How did you prepare these samples? Extrusion of films or casting? If the samples were produced by a different technology than samples used for mechanical and thermal properties you have to describe which one and mention process parameter. The used technology and its process parameters influence mechanical, and thermal properties as well biodegradation rate. Do you think that you could make any valuable statements about the influence of crystallinity degree on disintegration when comparing different samples? The different thicknesses and used technology will influence structure order?

Point 11: For biodegradation/ disintegration is crucial factor pH of compost as well as humidity. What was the pH of compost? Which compost did you use? Did you buy some commercial one? How did you achieve uniform level of humidity at 58°C within the experiment? Did you steer compost weekly? Did you measure humidity within the experiment?          

Point 12: The evaluation of results in chapter 3.2 is not sufficient. You should describe the results more and say some theory or statement. The highest drop of tensile strength is evident at 2.5 phr concentration of EBNO from shown results. Higher concentration did not evoke dramatic changes. Contrary, the elongation at break is continuously increasing with increasing concertation of EBNO. However, at 10 phr concertations of EBNO is evident an enormous drop of elongation at break. Do you have some explanation why? You compare achieved results with Sempere-Torregrosa et al., but you do not mention reference (line 269). The reference should be shown first (Sempere-Torregrosa et al. [x]). Compare all results with this Autor, not only elongation at break. Add a comparison of your results with other research works.  

Point 13: If you check the standard deviation of Young modulus and hardness in chapter 3.2 you cannot conclude any difference in results. Check and correct your statement.

Point 14. How can you explain decrease of impact strength of PLA at 5 and 7,5 phr concentration of plasticizer comparing to lower concertation when the elongation at break increase?

Point 15. How many samples did you measure in thermal analysis? Did you calculate the standard deviation? Where did you extract the samples (cross-section area)?  Explain it in the manuscript.    

Point 16: How can you explain that the highest decrease of initial decomposition temperature (T5%) is at PLA with 5 phr concertation?  Why do you not evaluate the decomposition temperature at 50% weight loss (T50%)? This result is used more often than Tmax. What causes the applied plasticizer? Compare the effect of concentration.

Point 17: You should mention that showed results of DSC analyse in table 2 and Figure 6 are from the second heating cycle (only material properties). You declare that used plasticizers evoke a decrease of Tg. However, the decrease is low. Is it the evaluation correct if you take into consideration the standard deviation that is missing?  Explain differences in Tm and discuss the influence of the used plasticizer on the crystallinity degree of PLA. Compare these results with other authors.   

Point 18: Which value of HDT result is correct for neat PLA? The value 59.2 °C is written in the text and in table it is 58.2°C. According to standard deviation, there are very small differences between evaluated results to state any valuable statements of the influence of plasticizers on HDT of PLA.

Point 19: The red and yellow arrow is not possible to see in the black and white version. Use another explanation (for example point 1). 

Point 20: Do you think that is evaluation of disintegration valuable after nine days? There are a lot of small pieces and a high increase in weight. How did you separate samples from compost? What was the standard deviation from 5 measurements? The samples after nine days of composting show differences. The PLA with 2.5 and 5 phr concentration of EBNO is disintegrated into smaller pieces compared to neat PLA and PLA+10 phr EBNO. Could it mean faster disintegration? Why? Could the plasticizer increase the hydrolysis of PLA?      

Round 2

Reviewer 2 Report

I accept the explanation of comments (points) 14, 16 and 18. However, I recommend you incorporate the discussion into the manuscript. For example, in point 18 you agree that the differences in HDT are very low. Nevertheless, the manuscript is unchanged, without any explanation.  

The measurement of disintegration on a single sample is very informative due to the potential influence of mechanical failure during the separation of the compost from PLA (Point 20).  I recommend focusing future work on a more relevant analysis of disintegration/biodegradation.  I would be very careful to predict the same structural properties for different technology and different specimens. One of the most critical parameters is for crystallization kinetics the thermodynamic conditions of solidification. Since the test specimens 1A type are 4 mm thick, cooled by a 25°C mold and films are 1mm thick (cooled by the ambient condition?), the influence of plasticizer on crystallinity degree could be different. 
